# Effects of Nitrogen Fertilizer on Photosynthetic Characteristics, Biomass, and Yield of Wheat under Different Shading Conditions

**Hongzhi Zhang [1,2,†], Qi Zhao [2,†], Zhong Wang [2,†], Lihong Wang [2], Xiaorong Li [3], Zheru Fan [2], Yueqiang Zhang [2], Jianfeng Li [2], Xin Gao [2], Jia Shi [2] and Fu Chen [1,\*]**

[1] College of Agronomy and Biotechnology, China Agricultural University/Key Laboratory of Farming System of Ministry of Agriculture and Rural Affairs, Beijing 100193, China; dreamzhz@xaas.ac.cn

[2] Institute of Nuclear and Biological Technologies, Xinjiang Academy of Agricultural Sciences/Key Laboratory of Oasis-Desert Crop Physiology Ecology and Cultivation of Ministry of Agricultural and Rural Affairs/Crop Chemical Regulation Engineering Technology Research Center in Xinjiang, Urumqi 830091, China; zhaoqi@xaas.ac.cn (Q.Z.); zhongwang@xaas.ac.cn (Z.W.); lihongwang@xaas.ac.cn (L.W.); zherufan@xaas.ac.cn (Z.F.); yueqiangzhang@xaas.ac.cn (Y.Z.); hssljf@xaas.ac.cn (J.L.); gaoxin0564@xaas.ac.cn (X.G.); Shij@xaas.ac.cn (J.S.)

[3] Institute of Nuclear and Biological Technologies, Xinjiang Academy of Agricultural Sciences/Xinjiang Key Laboratory of Crop Biotechnology, Urumqi 830091, China; lixiaorong@xaas.ac.cn

\* Correspondence: chenfu@cau.edu.cn; Tel./Fax: +86-1062733316

† These authors contributed equally to this work and are co-first authors.

**Abstract:** Fruit-wheat intercropping is an important way to resolve the land competition between fruit and grain and ensure food security. However, there is little research on the mechanism of wheat yield formation and its response to nitrogen fertilizer under long-term shading. From 2016 to 2017, wheat variety "Xindong 20" was selected, and four shading treatments were set: shading at jointing stage 10%-shading at heading stage 25% (S1), 20%–50% (S2), 30%–75% (S3), normal light (S0) and four nitrogen fertilizer (N0: 0 kg ha$^{-1}$, N1: 103.5 kg ha$^{-1}$, N2: 138 kg ha$^{-1}$, N3: 172.5 kg ha$^{-1}$). The results show that compared with S0, wheat leaf area index (LAI), chlorophyll a, b and a + b content under S1 increase by 14.9–57.4%, 2.9–24.5%, 16.5–28.9%, 7.8–25.5%, respectively, and they decrease significantly under S2 and S3. With the increase in the shading range, the net photosynthesis rate (Pn), transpiration rate (Tr), stomatal conductance (gs), and non-photochemical quentum coefficient (NPQ) decrease significantly, while the actual photochemical efficiency (ΦPSII) and the photochemical quenching coefficient (qP) increase significantly. Under S1, S2, and S3, the total dry matter accumulation (TDA), the dry matter accumulation of reproductive organs (DAR), and the yield decrease with the increase in shading range. Under the S0 and S1 conditions, compared with other nitrogen treatments, LAI, chlorophyll content, Pn, ΦPSII, qP, TDA, DAR, and yield of wheat under N2 treatment increase by 4.1–366.9%, 5.7–56.3%, 3.0–131.7%, 6.7–87.5%, 3.7–96.9%, 7.1–340.8%, 0.3–323.0%, 1.5–231.2%. Therefore, under jujube-wheat intercropping, and apricot-wheat and walnut-wheat with light shade in the early stage, photosynthetic capacity of wheat leaves and dry matter accumulation and transfer to grains can be regulated by proper nitrogen application, which is beneficial to compensate for the negative effects of insufficient light on wheat yield; under moderate or excessive shading conditions (apricot-wheat and walnut-wheat in full fruit period), the regulating effect of nitrogen application on wheat is reduced, and the nitrogen application should be moderately reduced.

**Keywords:** shading; nitrogen application; photosynthetic characteristics; yield; wheat

## 1. Introduction

Wheat (*Triticum aestivum* L.) is one of the most important food crops in China, and its output is restricted by varieties and external environment. Intercropping of trees and food crops has been practiced in China and other countries for thousands of years [1,2]. In recent

decades, researchers in many fields have become increasingly interested in traditional agroforestry systems because of their potential to increase land productivity, diversify production, and improve resource utilization [3–6]. In China, forest-grain intercropping has been implemented for hundreds of years, which is an important measure to ensure food security [4,5]. However, in most forest intercropping systems, light competition between trees and crops is the main reason for the decline in crop yields [6].

Light and nitrogen are two important factors that affect crop growth. Light is a prerequisite for photosynthesis and growth of plants. Nitrogen nutrition directly affects the photosynthetic rate and growth, and ultimately affects yield and light energy utilization. Radiation changes affect the efficiency of light and carbon use, and ultimately affect total food production [7–10]. Shading can significantly impair the net photosynthesis of wheat leaves through changing chloroplast function [11] and inhibiting photosystem II activity (PSII) at any growth stage [12–15]. Increasing application of nitrogen fertilizer is beneficial to enhancing the ability of wheat leaves to capture light energy, increasing the conversion efficiency of light energy and the ratio of the open part of the PSII, reducing the heat dissipation of non-radiative energy, and facilitating wheat to use the captured light energy more effectively for photosynthesis [16], and then increasing Pn. Under shading condition, increasing nitrogen fertilizer is helpful to increase chlorophyll content and photosynthetic rate of leaves, but the effect of nitrogen fertilizer on increasing yield is much lower than that under normal light [17]. The kernels per ear and yield increase after nitrogen application under normal light, but the kernels per ear decreases when nitrogen is applied under shading conditions. Studies on the interaction effects of light and nitrogen mainly focus on corn, rice, vegetables and tobacco [18–20]. There are few studies on the effect of nitrogen fertilizer on the photosynthetic physiological mechanism of wheat such as light energy absorption, transmission, dissipation and distribution under long-term shading conditions.

The regional advantages of characteristic fruit trees (walnuts, apricots and jujubes) in southern Xinjiang region play a pivotal role in the fruit industry in Xinjiang. There are more people and less land in the three southern prefectures of Xinjiang, and the per capita cultivated land is less than 0.16 ha [21]. In this region, more than 80% of fruit trees are intercropped with fruit farmers, and about 70% of field crops such as grain and cotton are intercropped with fruit trees, the development of characteristic forests and fruits has played an important role in the development of local rural economy and the increase in farmers' income [22]. However, with the expansion of the farmland area, the age of the fruit trees and the expansion of the canopy, the shade of the wheat by the fruit trees has increased, resulting in a decrease in the grain weight and yield of wheat [23,24]. Nitrogen fertilizer is an important measure to regulate crop yield formation [25,26]. The regulating effect of nitrogen fertilizer on photosynthetic characteristics, dry matter accumulation and yield of wheat under different shading conditions is an issue that needs to be further explored. Hence, in order to have a comprehensive understanding of fruit and wheat intercropping, in this study, during the transition period between fruit trees and wheat phenology (from wheat jointing stage to mature stage), shading is set up artificially to simulate the different degrees of shading by fruit trees at the middle and late stages of wheat growth. Then the regulating effects of nitrogen fertilizer on wheat leaf area, chlorophyll content, gas exchange parameters, and chlorophyll fluorescence parameters under different shading conditions are explored to study the relationship between photosynthetic characteristics and dry matter accumulation and yield of wheat. It is expected to provide a theoretical basis for high-yield wheat cultivation and nitrogen management under long-term shading in the fruit tree-wheat intercropping model.

## 2. Materials and Methods

### 2.1. Experimental Site and Cultivar

The experiment was carried out with the velvet plant of Xinjiang Zepu County Seed Company (Kashi Prefecture, Xinjiang Uygur Autonomous Region, China.) in 2016–2017 (E: 77°16′, N: 38°10′), at an altitude of 1266 m. Annual mean temperature is 11.6 °C (1961–

2008). Cumulative temperatures above 0 °C is 4183 °C. The mean frost-free period is 212 days. Annual precipitation is 54.8 mm, potential evaporation is 2079 mm, and the region has a typical arid climate. The soil at the site is classified as an arenosol in the classification system of the Food and Agriculture Organization (FAO), and Table 1 lists the local soil chemical properties (Agricultural Product Quality Inspection Center, Ministry of Agriculture (Urumqi)). The tested winter wheat variety was Xindong 20. This cultivar was developed by the Xinjiang Academy of Agricultural Sciences and Reclamation Science and is officially registered and released by the Xinjiang Crop Cultivar Registration Committee. The total growth period from emergence to mature period is 238 days, semi-winterness variety.

**Table 1.** Soil conditions of the trial plots.

| Layer (cm) | Organic Substance (g kg$^{-1}$) | Total Nitrogen (g kg$^{-1}$) | Total Phosphorus (g kg$^{-1}$) | Total Potassium (g kg$^{-1}$) | Available Nitrogen (mg kg$^{-1}$) | Available Phosphorus (mg kg$^{-1}$) | Available Potassium (mg kg$^{-1}$) | Total Salt (g kg$^{-1}$) | pH |
|---|---|---|---|---|---|---|---|---|---|
| 0–20 | 14.87 | 0.635 | 0.796 | 19.019 | 37.3 | 18.2 | 104 | 1.4 | 8.51 |
| 20–40 | 14.13 | 0.464 | 0.668 | 19.534 | 33.7 | 4.6 | 94 | 0.5 | 8.92 |

*2.2. Experiment Design*

The split-plot design was adopted in the experiment, with shading treatment as the main plot and nitrogen fertilizer treatment as the secondary plot. There were 4 shading levels, S1: jointing stage shading 10%-heading stage shading 25%, S2: jointing stage shading 20%-heading stage shading 50%, S3: jointing stage shading 30%-heading stage shading 75%. S0: shading 0%, the shading treatments were conducted until to mature stage. There were four nitrogen fertilizer treatments levels, N0: no fertilizer during the entire growth period, N1: at jointing stage (after shading), applied nitrogen 103.5 kg ha$^{-1}$, N2: at jointing stage (after shading) applied nitrogen 138 kg ha$^{-1}$ (conventional), N3: at the jointing stage (after shading) applied nitrogen 172.5 kg ha$^{-1}$. The plot area was 8 m$^2$, and the plot was repeated three times.

Black nylon net with different light transmittance was used for shade, and the shade net was kept about 50 cm away from the surface of wheat canopy (adjusted with the height of wheat plants) to ensure good ventilation in the colony. Shading area exceeded the four sides of the area, and the treatment area was completely covered. During the nitrogen fertilizer treatment, the guard row was used to separate to prevent water content and nutrient side infiltration. The seeds were sown on 30 September 2015 and 3 October 2016, respectively. Before sowing wheat, 150 kg ha$^{-1}$ of urea (comprising 46% N) and 375 kg ha$^{-1}$ of diammonium phosphate (with 46% $P_2O_5$, 18% N) were applied as a base fertilizer, and 150 kg ha$^{-1}$ of urea was applied during the greening period (N0 treatment: no fertilizer throughout the growth period). The winter wheat was sown 270 kg ha$^{-1}$, with a row spacing of 20 cm. It was irrigated for six times during the whole growth period (overwintering, greening, jointing, booting, flowering, and filling), and the total amount of irrigation was 3600 m$^3$. The drip irrigation was adopted, and the water quantity was controlled by a water meter and a ball valve. Other cultivation measures were consistent for all treatments.

*2.3. Measurement Items and Methods*

2.3.1. LAI

At the flowering stage and filling stage, 10 representative plants were continuously selected from each treatment to measure the length and width of green leaves of each plant and calculate the leaf area (length × width × 0.83). Meanwhile, LAI was calculated according to the total stem number in each growth stage.

### 2.3.2. Chlorophyll Content Measurement

According to the method of Hesheng Li [27], take the flag leaves of wheat, wipe them clean, remove the main veins, cut them into pieces and mix them well. Then, 0.2 g was weighed, and it was ground with 95% ethanol, then filtered and diluted to 25 mL. OD values were measured at 665, 649 and 470 nm wavelengths by spectrophotometer, and 95% ethanol was used as blank control. The calculation formula is as follows:

$$\rho(Chla) = 13.95\ D_{665} - 6.88\ D_{649} \tag{1}$$

$$\rho(Chlb) = 24.96\ D_{649} - 7.32\ D_{665} \tag{2}$$

$\rho(Chla)$ and $\rho(Chlb)$: pigment concentration (mg mL$^{-1}$); $D_{665}$, $D_{649}$ and $D_{470}$: the OD value was measured at wavelengths of 663 nm, 649 nm and 470 nm; the content of pigment in the leaves (mg g$^{-1}$) = pigment concentration $\times$ extract veolume $\times$ dilution factor/fresh weight of the sample.

### 2.3.3. Measurement of Blade Gas Exchange Parameters

Gas exchange (Pn, gs, Tr and Ci) in wheat flag leaves was determined by li-6400 portable photosynthesis system (LI-6400, Li-COR INC., NE, USA). For each treatment, 5 to 6 leaves were selected, red and blue light sources (LED) were used, open air path was used, air flow rate was 500 $\mu$mol$\cdot$s$^{-1}$, and light intensity was set according to light intensity, different shade range and temperature and humidity at 10:30 to 13:00 under natural environmental conditions in Xinjiang. S0: 1800 $\mu$mol m$^{-2}$ s$^{-1}$, S1: 1350 $\mu$mol m$^{-2}$ s$^{-1}$, S2: 900 $\mu$mol m$^{-2}$ s$^{-1}$, S3: 450 $\mu$mol m$^{-2}$ s$^{-1}$. The temperature remained at 32, the $CO_2$ concentration was basically kept at about 380 $\mu$mol mol$^{-1}$, and the relative humidity was 30%–32%.

### 2.3.4. Chlorophyll Fluorescence Measurement

The chlorophyll fluorescence parameters of flag leaves during flowering and filling stages were measured with a pulse-modulated fluorometer FMS-2 (FMS-2, Hansatech Instruments Led., Norfolk, UK). Fluorescence parameters Fs and Fm were measured under light (S0 treatment: light intensity was set as 1800 $\mu$mol m$^{-2}$ s$^{-1}$, S1 treatment: 1350 $\mu$mol m$^{-2}$ s$^{-1}$, S2 treatment: 900 $\mu$mol m$^{-2}$ s$^{-1}$, S3 treatment: 450 $\mu$mol m$^{-2}$ s$^{-1}$), and F0 was measured by far red light. Then F0, Fm, Fv/Fm and other parameters were measured under dark adaption for 15 min. $\Phi$PSII, qP, and NPQ were calculated according to the relevant formula. In all, 5–8 leaves were measured in each plot.

### 2.3.5. Dry Matter Accumulation and Distribution

At flowering stage, filling stage and maturity stage, 20 plants were taken from each plot and divided into stem, leaf, ear (grain and glume at maturity stage). The plants were finished at 105 °C for 30 min, dried at 80 °C and weighed, and the accumulation and distribution of photosynthetic matter were measured.

### 2.3.6. Yield and Its Components

At the mature stage, a representative 2 rows of 2 m indoor seed test was taken from each plot to calculate the grains per hectare, kernels per year and 1000-grain weight. The 4 m$^2$ of each plot was harvested for yield determination.

### *2.4. Data Analysis*

SPSS 16.0 software (SPSS Institute Inc., Chicago, IL, USA) was used to analyze the data, Duncan's new multiple-range test was used to test the difference at the level of 0.05. Plot with Sigmaplot 12.5 (Aspire Software Intl., Ashburn, VA, USA).

## 3. Result and Analysis

### 3.1. LAI

The experiment showed (Figure 1) that LAI at the flowering stage and filling stage both increased first and then decreased with the increase in the shading range. LAI of S1 treatment was significantly higher than other treatments at flowering stage. Compared with S0, S2 and S3 treatments, S1 treatment increased by 14.9–57.4%, 10.8–54.4% and 36.2–117.8%. There was no obvious difference in LAI between S1 and S2 treatments at filling stage, but the LAI of S1 and S2 treatments was significantly higher than that of S0 and S3 treatments. Under the conditions of S0 and S1, LAI first increased and then decreased with the increase in nitrogen application. The LAI of N2 treatment was the highest, and it was 11.8–50.1%, 4.1–73.6% and 78.1–366.9% higher than that of N1, N3 and N0 treatments, respectively; under the conditions of S2 and S3, there was no significant difference in the LAI among different nitrogen treatment at the flowering stage. The LAI at filling stage decreased with the increase in the nitrogen application, and the LAI of N1 treatment was the highest.

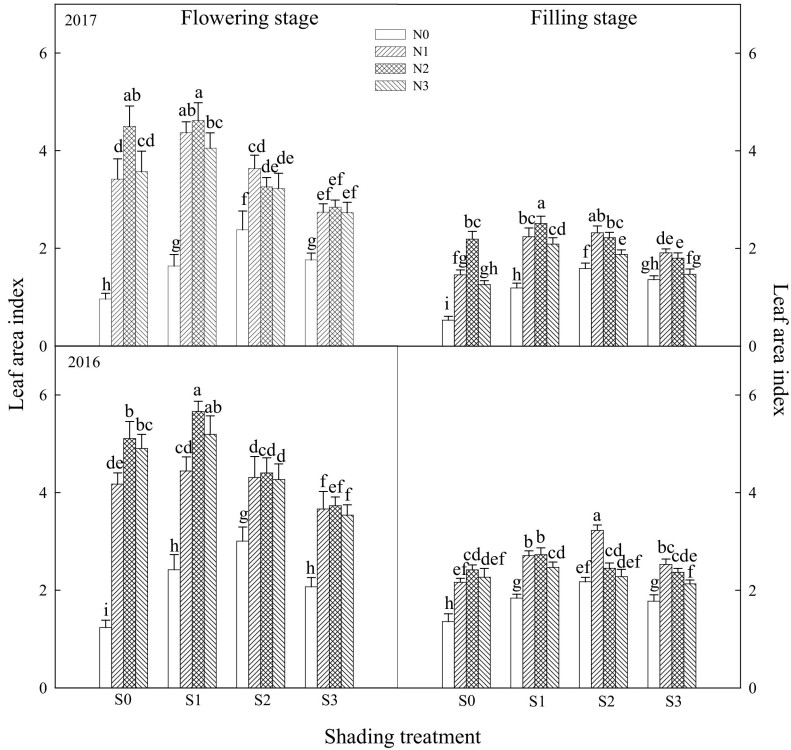

**Figure 1.** Effect of light and nitrogen treatment on the leaf area index (LAI) of wheat. Bars indicate SD (n = 3). Values within columns followed by the same letter are statistically insignificant at the 0.05 level.

### 3.2. Chlorophyll

The experiment showed (Table 2) that with the increase in the shade range, the chlorophyll a, b, a + b increased first and then decreased, and the chlorophyll of S1 treatment was the highest. At flowering stage and filling stage, the chlorophyll of S1 treatment was 2.9%, 5.1%, 6.3% and 24.5%, 10.7%, 11.8% higher than that of S0, S2, and S3, respectively; chlorophyll b was 16.5%, 4.6%, 9.6%, and 28.9%, 3.55%, 0.6% higher than that of S0, S2, and S3, respectively; chlorophyll a + b of S1 treatment was 7.75%, 4.9%, 7.4%, and 25.5%, 7.2%, 6.9% higher than that of S0, S2, and S3; chlorophyll a/b decreased. Under the conditions of S0 and S1, the content of chlorophyll a, b, and a + b first increased and then decreased with the increase in nitrogen application. The chlorophyll of N2 treatment was the highest. Chlorophyll a of N2 treatment was 11.6%, 5.0%, 3.0% higher than that of N0,

N1, and N3 treatments on average at flowering stage, and chlorophyll a of N2 treatment was 49.9%, 9.2%, and 7.7% higher than that of N0, N1, and N3 treatment at filling stage; chlorophyll b of N2 treatment was 45.3%, 23.6%, 15.1% higher than that of N0, N1, and N3 treatment at flowering stage, and chlorophyll b of N2 treatment was 64.9%, 10.7%, and 7.1% higher than that of N0, N1, and N3 treatment at filling stage; chlorophyll a + b of N2 treatment was 20.8%, 11.0%, 5.7% higher than that of N0, N1, and N3 treatment at flowering stage, and chlorophyll a + b of N2 treatment was 56.3%, 8.9%, 7.0% higher than that of N0, N1, and N3 treatment at filling stage; under the conditions of S2 and S3, chlorophyll a, b, and a + b decreased with the increase in nitrogen application; under different shading conditions, the chlorophyll a/b without nitrogen application (N0) was higher than that with nitrogen application. There was no significant difference in chlorophyll a/b among different nitrogen treatments.

### 3.3. Gas Exchange Parameters

The experiment showed (Figure 2) that the Pn decreased with the increase in the shading range. There was no significant difference in Pn between S0 and S1 treatments, but Pn of S0 treatment was significantly higher than that of S2 and S3 treatments. Pn of S2 and S3 treatments was 4.55–24.7% and 54.7–74.4% lower than that of S0 treatment, respectively. Under the conditions of S0 and S1 treatment, Pn increased first and then decreased with the increase in nitrogen application. The Pn of N2 treatment was significantly higher than other treatments. The Pn of N2 was 3.0–13.7%, 3.2–33.7% and 8.9–131.7% higher than that of N3, N1 and N0 treatments, respectively; under S2 and S3 conditions, the Pn of N1 treatment was the highest, there was no significant difference between different nitrogen treatments. The change trend of Tr and gs was basically consistent with that of Pn (Figures 3 and 4).

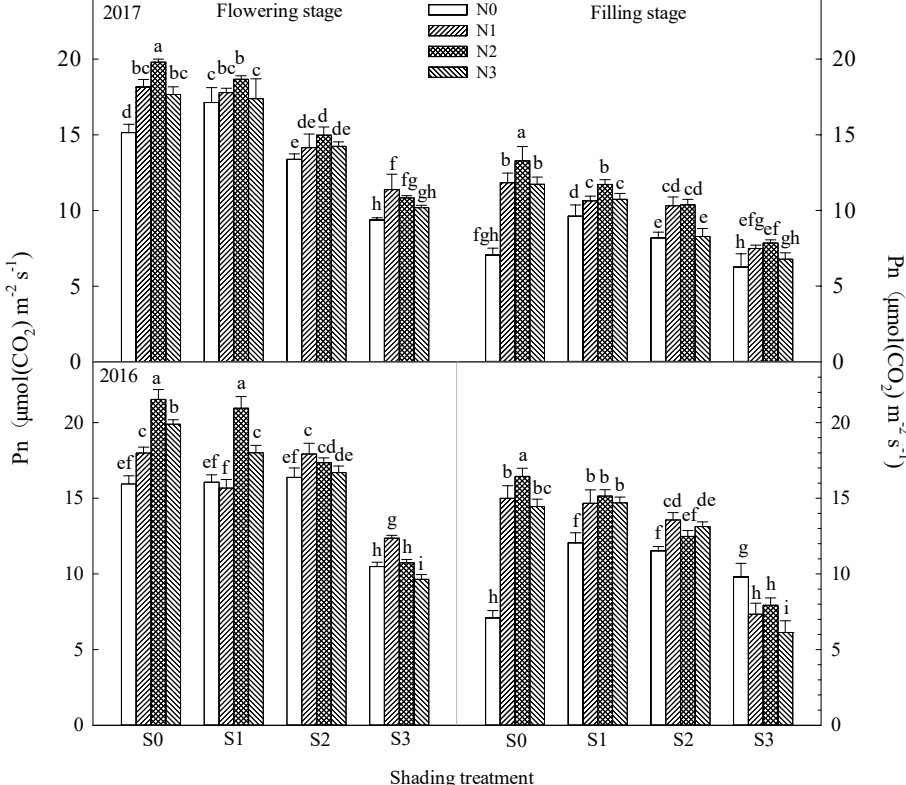

**Figure 2.** Effect of light and nitrogen treatment on the net photosynthetic rate (Pn) of wheat. Bars indicate SD (n = 3). Values within columns followed by the same letter are statistically insignificant at the 0.05 level.

**Table 2.** Effect of light and nitrogen treatment on the chlorophyll Chl a, Chl b, Chl (a + b), and the Chl a/b ratio of wheat.

| Treatment | | CHl a | | | | CHl b | | | | CHl (a + b) | | | | Chl a/b | | | |
|---|---|---|---|---|---|---|---|---|---|---|---|---|---|---|---|---|---|
| | | 2017 | | 2016 | | 2017 | | 2016 | | 2017 | | 2016 | | 2017 | | 2016 | |
| | | Flowering Stage | Filling Stage | Flowering Stage | Filling Stage | Flowering Stage | Filling Stage | Flowering Stage | Filling Stage | Flowering Stage | Filling Stage | Flowering Stage | Filling Stage | Flowering Stage | Filling Stage | Flowering Stage | Filling Stage |
| S0 | N0 | 2.18 ± 0.10 [j] | 1.18 ± 0.01 [d] | 2.31 ± 0.04 [e] | 1.35 ± 0.03 [g] | 1.04 ± 0.06 [g] | 0.35 ± 0.01 [f] | 1.02 ± 0.03 [h] | 0.42 ± 0.02 [g] | 3.22 ± 0.01 [g] | 1.54 ± 0.04 [e] | 3.33 ± 0.11 [f] | 1.78 ± 0.03 [j] | 2.00 ± 0.12 [a] | 3.29 ± 0.01 [a] | 2.27 ± 0.03 [a] | 3.14 ± 0.03 [a] |
| | N1 | 2.37 ± 0.06 [efg] | 2.02 ± 0.05 [bc] | 2.47 ± 0.12 [bcd] | 2.01 ± 0.08 [ef] | 1.28 ± 0.06 [cde] | 0.76 ± 0.04 [cde] | 1.27 ± 0.02 [g] | 0.81 ± 0.07 [e] | 3.65 ± 0.02 [cde] | 2.78 ± 0.09 [cd] | 3.74 ± 0.09 [de] | 2.83 ± 0.06 [h] | 1.84 ± 0.07 [cde] | 2.66 ± 0.10 [cd] | 1.93 ± 0.08 [b] | 2.46 ± 0.08 [bc] |
| | N2 | 2.52 ± 0.13 [b] | 2.06 ± 0.04 [bc] | 2.56 ± 0.05 [ab] | 2.34 ± 0.08 [ab] | 1.41 ± 0.12 [ab] | 0.76 ± 0.03 [cde] | 1.74 ± 0.05 [bc] | 0.95 ± 0.02 [bcd] | 3.93 ± 0.15 [ab] | 2.83 ± 0.07 [cd] | 4.31 ± 0.13 [ab] | 3.28 ± 0.05 [bc] | 1.78 ± 0.02 [cdefg] | 2.71 ± 0.09 [bc] | 1.47 ± 0.09 [efg] | 2.46 ± 0.08 [bc] |
| | N3 | 2.44 ± 0.09 [cd] | 1.91 ± 0.09 [c] | 2.53 ± 0.05 [abc] | 2.09 ± 0.03 [de] | 1.14 ± 0.07 [fg] | 0.73 ± 0.02 [de] | 1.57 ± 0.12 [e] | 0.84 ± 0.02 [de] | 3.58 ± 0.03 [def] | 2.64 ± 0.04 [d] | 4.11 ± 0.04 [bc] | 2.93 ± 0.09 [fgh] | 2.12 ± 0.02 [a] | 2.59 ± 0.07 [cd] | 1.61 ± 0.11 [de] | 2.47 ± 0.05 [bc] |
| S1 | N0 | 2.40 ± 0.04 [def] | 2.03 ± 0.07 [bc] | 2.41 ± 0.09 [bcde] | 1.90 ± 0.03 [f] | 1.25 ± 0.02 [ef] | 0.69 ± 0.06 [e] | 1.45 ± 0.06 [f] | 0.61 ± 0.02 [f] | 3.64 ± 0.01 [cde] | 2.71 ± 0.03 [cd] | 3.87 ± 0.09 [cd] | 2.51 ± 0.05 [i] | 1.92 ± 0.09 [bc] | 2.93 ± 0.11 [b] | 1.66 ± 0.15 [cd] | 3.12 ± 0.13 [a] |
| | N1 | 2.46 ± 0.01 [cd] | 2.46 ± 0.04 [a] | 2.51 ± 0.18 [abc] | 2.39 ± 0.06 [ab] | 1.36 ± 0.01 [bc] | 0.96 ± 0.02 [ab] | 1.58 ± 0.04 [de] | 0.99 ± 0.07 [abc] | 3.82 ± 0.12 [bcd] | 3.42 ± 0.10 [a] | 4.09 ± 0.21 [bc] | 3.38 ± 0.11 [ab] | 1.79 ± 0.07 [cdefg] | 2.54 ± 0.02 [cde] | 1.59 ± 0.02 [def] | 2.39 ± 0.06 [c] |
| | N2 | 2.58 ± 0.06 [a] | 2.54 ± 0.03 [a] | 2.62 ± 0.12 [a] | 2.44 ± 0.05 [a] | 1.50 ± 0.03 [a] | 0.94 ± 0.06 [ab] | 1.81 ± 0.04 [ab] | 1.03 ± 0.12 [ab] | 4.08 ± 0.24 [a] | 3.49 ± 0.01 [a] | 4.44 ± 0.11 [a] | 3.47 ± 0.15 [a] | 1.71 ± 0.09 [efg] | 2.68 ± 0.15 [c] | 1.45 ± 0.06 [efg] | 2.35 ± 0.05 [c] |
| | N3 | 2.47 ± 0.08 [bc] | 2.45 ± 0.10 [a] | 2.50 ± 0.03 [abc] | 2.36 ± 0.09 [ab] | 1.41 ± 0.12 [ab] | 0.97 ± 0.02 [ab] | 1.86 ± 0.06 [a] | 1.02 ± 0.07 [ab] | 3.88 ± 0.02 [abc] | 3.42 ± 0.05 [a] | 4.37 ± 0.19 [a] | 3.38 ± 0.02 [ab] | 1.74 ± 0.12 [defg] | 2.50 ± 0.02 [cde] | 1.34 ± 0.12 [g] | 2.31 ± 0.09 [cd] |
| S2 | N0 | 2.26 ± 0.03 [i] | 2.33 ± 0.23 [ab] | 2.27 ± 0.05 [e] | 2.12 ± 0.05 [de] | 1.10 ± 0.01 [g] | 0.92 ± 0.08 [b] | 1.27 ± 0.03 [g] | 0.87 ± 0.01 [de] | 3.36 ± 0.02 [fg] | 3.26 ± 0.11 [ab] | 3.55 ± 0.05 [ef] | 2.99 ± 0.14 [fg] | 2.04 ± 0.01 [ab] | 2.50 ± 0.12 [cde] | 1.79 ± 0.12 [bc] | 2.43 ± 0.05 [c] |
| | N1 | 2.36 ± 0.02 [efg] | 2.08 ± 0.02 [bc] | 2.50 ± 0.02 [abc] | 2.25 ± 0.11 [bcd] | 1.41 ± 0.04 [ab] | 0.95 ± 0.02 [ab] | 1.70 ± 0.04 [bcd] | 1.04 ± 0.10 [ab] | 3.77 ± 0.02 [bcde] | 3.03 ± 0.09 [bc] | 4.21 ± 0.07 [ab] | 3.29 ± 0.06 [bc] | 1.67 ± 0.02 [g] | 2.19 ± 0.02 [fg] | 1.47 ± 0.05 [efg] | 2.15 ± 0.11 [de] |
| | N2 | 2.34 ± 0.07 [fgh] | 1.94 ± 0.56 [c] | 2.51 ± 0.04 [abc] | 2.16 ± 0.14 [cde] | 1.38 ± 0.04 [bc] | 0.76 ± 0.03 [cde] | 1.72 ± 0.11 [bc] | 0.93 ± 0.09 [bcd] | 3.73 ± 0.01 [bcde] | 2.69 ± 0.32 [cd] | 4.24 ± 0.13 [ab] | 3.09 ± 0.12 [def] | 1.68 ± 0.15 [fg] | 2.53 ± 0.30 [cde] | 1.45 ± 0.09 [efg] | 2.30 ± 0.15 [cd] |
| | N3 | 2.41 ± 0.08 [de] | 1.89 ± 0.11 [c] | 2.34 ± 0.12 [de] | 2.01 ± 0.02 [ef] | 1.37 ± 0.08 [bc] | 0.81 ± 0.02 [c] | 1.73 ± 0.11 [bc] | 1.00 ± 0.07 [ab] | 3.78 ± 0.10 [bcde] | 2.70 ± 0.02 [cd] | 4.07 ± 0.20 [bc] | 3.02 ± 0.12 [efg] | 1.74 ± 0.03 [defg] | 2.32 ± 0.22 [ef] | 1.35 ± 0.05 [g] | 1.99 ± 0.02 [ef] |
| S3 | N0 | 2.29 ± 0.02 [hi] | 2.54 ± 0.09 [a] | 2.29 ± 0.06 [e] | 2.28 ± 0.14 [abc] | 1.12 ± 0.04 [g] | 1.01 ± 0.03 [a] | 1.21 ± 0.05 [g] | 0.89 ± 0.01 [cde] | 3.41 ± 0.09 [fg] | 3.55 ± 0.09 [a] | 3.50 ± 0.16 [f] | 3.17 ± 0.03 [cde] | 2.04 ± 0.01 [ab] | 2.51 ± 0.03 [cde] | 1.89 ± 0.05 [b] | 2.56 ± 0.1 [b] |
| | N1 | 2.36 ± 0.03 [efg] | 1.99 ± 0.32 [bc] | 2.51 ± 0.01 [abc] | 2.12 ± 0.06 [de] | 1.36 ± 0.03 [bcd] | 0.96 ± 0.05 [ab] | 1.72 ± 0.04 [bc] | 1.06 ± 0.06 [a] | 3.72 ± 0.07 [bcde] | 2.96 ± 0.23 [bcd] | 4.23 ± 0.08 [ab] | 3.19 ± 0.08 [cd] | 1.73 ± 0.02 [defg] | 2.07 ± 0.01 [g] | 1.46 ± 0.03 [efg] | 1.98 ± 0.06 [ef] |
| | N2 | 2.30 ± 0.05 [hi] | 1.89 ± 0.12 [c] | 2.40 ± 0.04 [cde] | 2.08 ± 0.12 [e] | 1.25 ± 0.07 [de] | 0.78 ± 0.05 [cd] | 1.67 ± 0.07 [cde] | 1.01 ± 0.04 [ab] | 3.55 ± 0.30 [ef] | 2.67 ± 0.02 [d] | 4.07 ± 0.14 [bc] | 3.09 ± 0.09 [def] | 1.83 ± 0.01 [cdef] | 2.42 ± 0.12 [def] | 1.43 ± 0.14 [fg] | 2.05 ± 0.12 [e] |
| | N3 | 2.32 ± 0.04 [gh] | 1.84 ± 0.09 [c] | 2.29 ± 0.03 [e] | 1.87 ± 0.12 [f] | 1.24 ± 0.03 [ef] | 0.79 ± 0.02 [cd] | 1.65 ± 0.11 [cde] | 0.99 ± 0.03 [abc] | 3.56 ± 0.22 [ef] | 2.63 ± 0.56 [d] | 3.94 ± 0.16 [cd] | 2.86 ± 0.05 [gh] | 1.87 ± 0.10 [cd] | 2.33 ± 0.24 [ef] | 1.38 ± 0.06 [g] | 1.86 ± 0.12 [f] |

Values are means ± SD (*n* = 3). Values within columns followed by the same letter are statistically insignificant at the 0.05 level.

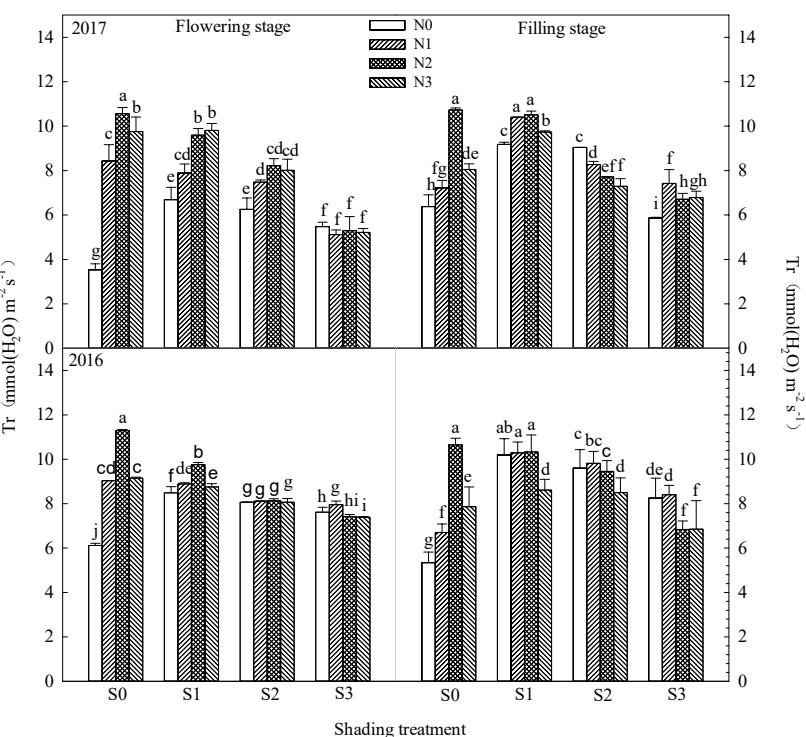

**Figure 3.** Effect of light and nitrogen treatment on the transpiration rate (Tr) of wheat. Bars indicate SD (n = 3). Values within columns followed by the same letter are statistically insignificant at the 0.05 level.

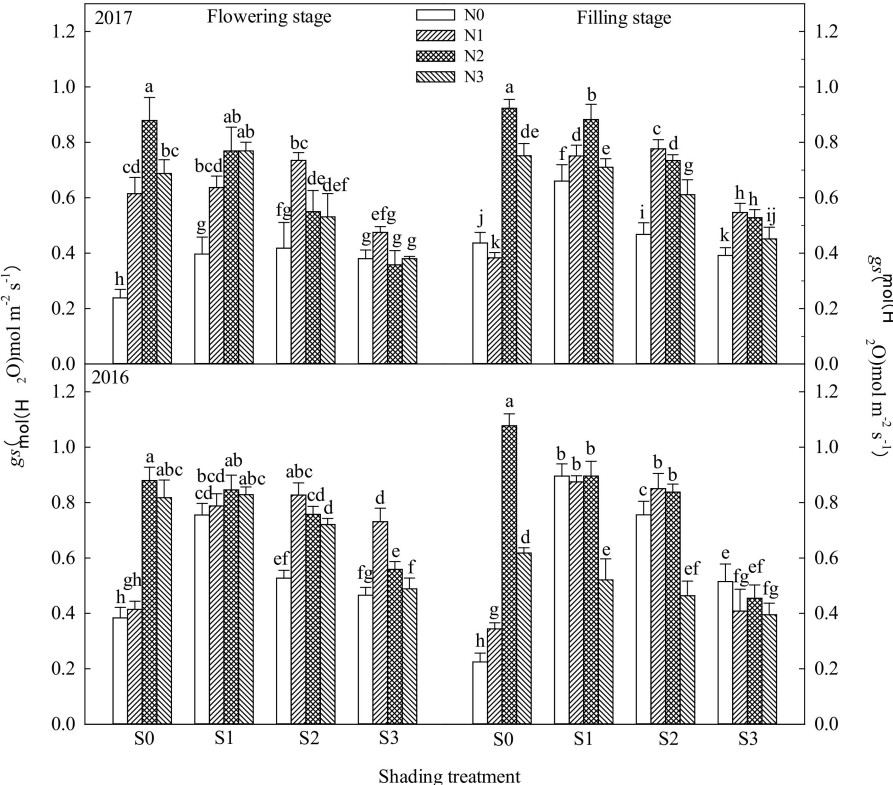

**Figure 4.** Effect of light and nitrogen treatment on the stomatal conductance (gs) of wheat. Bars indicate SD (n = 3). Values within columns followed by the same letter are statistically insignificant at the 0.05 level.

The Ci showed an increasing trend with the increase in the shading range (Figure 5). Under the conditions of S0 and S1, the Ci of N2 and N3 treatment was higher than that of N0 and N1 treatment; under the conditions of S2 and S3, Ci was not affected by nitrogen application.

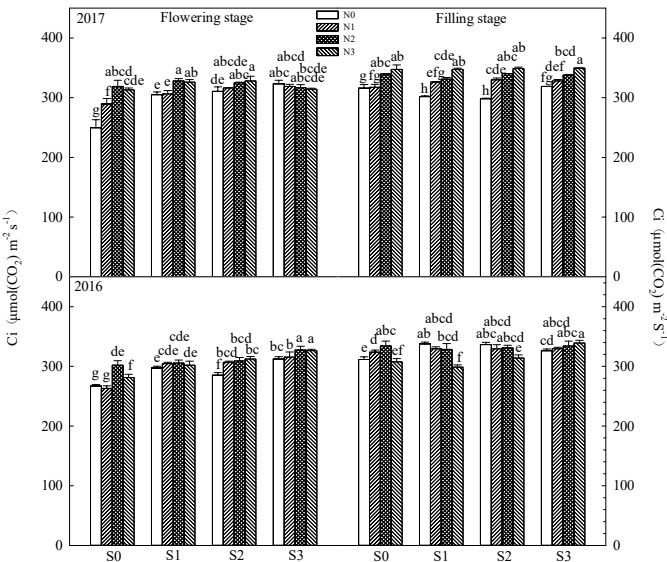

**Figure 5.** Effect of light and nitrogen treatment on the intercellular $CO_2$ concentration (Ci) of wheat. Bars indicate SD (n = 3). Values within columns followed by the same letter are statistically insignificant at the 0.05 level.

### 3.4. Chlorophyll Fluorescence Parameters

Fv/Fm reflects the maximum photochemical efficiency of the PSII. The experiment showed (Figure 6) that there was no significant difference in Fv/Fm among different treatments, indicating that shading and nitrogen application had no effect on the maximum photochemical efficiency of PS of wheat.

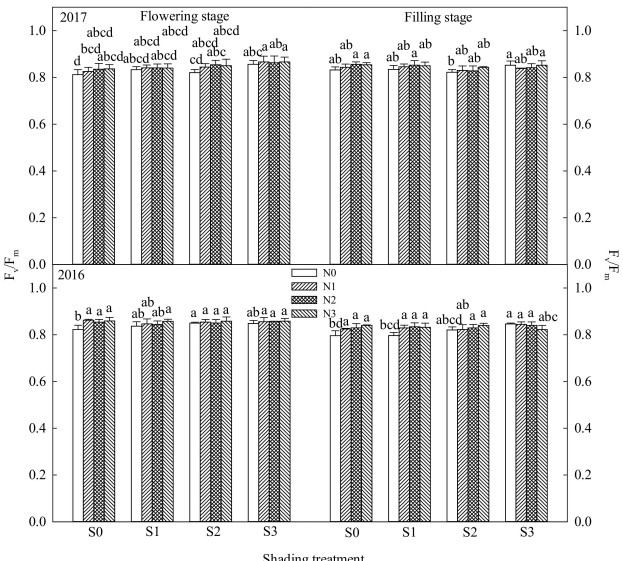

**Figure 6.** Effect of light and nitrogen treatment on maximum quantum efficiency of PSII photochemistry (Fv/Fm) of wheat. Bars indicate SD (n = 3). Values within columns followed by the same letter are statistically in-significant at the 0.05 level.

Figure 7 showed the effect of light and nitrogen on the quantum efficiency of PSII (ΦPSII) of wheat. It shows that with the increase in the shade range, ΦPSII increased significantly, and the ΦPSII of the S1, S2, and S3 treatments were 10.8–50.3%, 33.2–111.4%, 54.2–180.4% higher than that of S0 treatment, respectively. Under different shading conditions, ΦPSII first increased and then decreased with the increase in nitrogen application. Under the conditions of S0 and S1, the ΦPSII of N2 treatment was significantly higher than other treatments, which was 6.7–22.3%, 0.8–46.0%, 38.9–87.5% higher than that of N3, N1, and N0, respectively; under S2 and S3 conditions, the ΦPSII of N1 treatment was the highest.

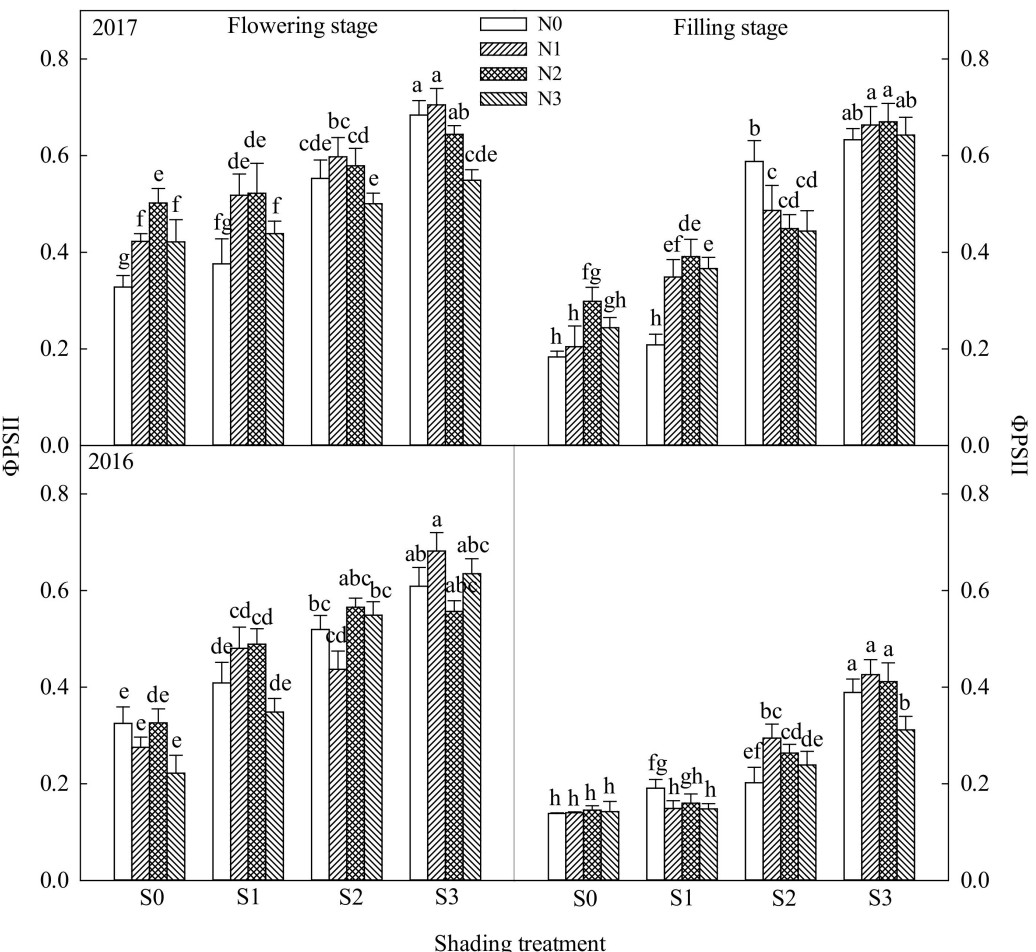

**Figure 7.** Effect of light and nitrogen treatment on the quantum efficiency of PSII (ΦPSII) of wheat. Bars indicate SD (n = 3). Values within columns followed by the same letter are statistically insignificant at the 0.05 level.

The qP reflects the efficiency of the light quantum captured by PSII converted into chemical energy, and it represents the openness of the PSII reaction center. The larger the qP, the larger the electron transfer activity of PSII. The changing trend of the qP is basically consistent with that of ΦPSII (Figure 8).

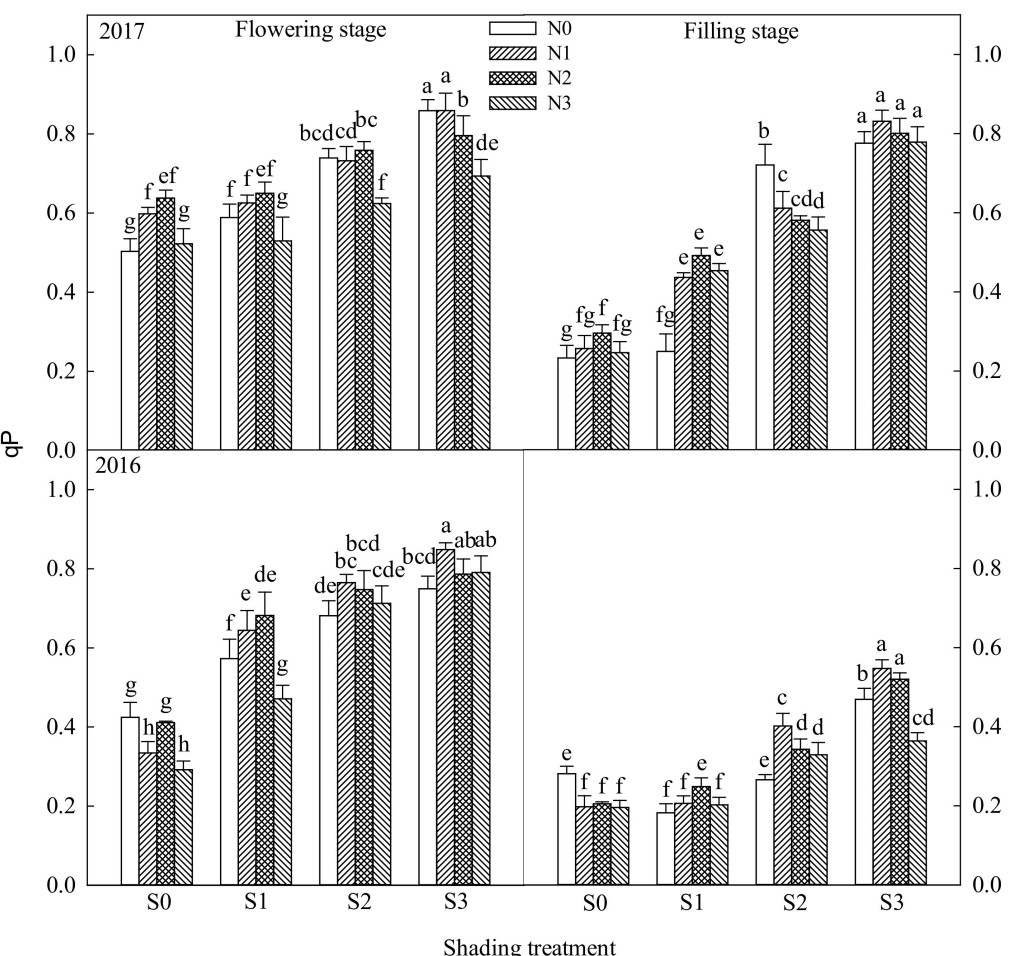

**Figure 8.** Effect of light and nitrogen treatment on the photochemical quenching coefficient (qP) of wheat. Bars indicate SD (n = 3). Values within columns followed by the same letter are statistically insignificant at the 0.05 level.

The NPQ reflects the non-radiative energy dissipation of the PSII reaction center. The experiment showed (Figure 9) that at flowering stage, NPQ significantly decreased with the increase in shading range, and the NPQ of S1, S2, and S3 treatments were 21.8–23.5%, 40.2–41.3%, 53.6–60.9%, lower that of S0 treatment, respectively; at filing stage, the NPQ first increased and then decreased, and the NPQ of S1 treatment was significantly higher than other treatments, which was 16.3–72.0% higher than that of other treatments. Under S0 and S1 conditions, the NPQ of N2 treatment was the lowest, which was 2.2–20.0% and 9.6–29.1% lower than that of N1 and N3 treatments, respectively; under the conditions of S2 and S3, NPQ increased with the increase in nitrogen application, and the NPQ of N2 and N3 treatments were 5.5–71.6% and 14.0–75.5% higher than that of N1 treatments.

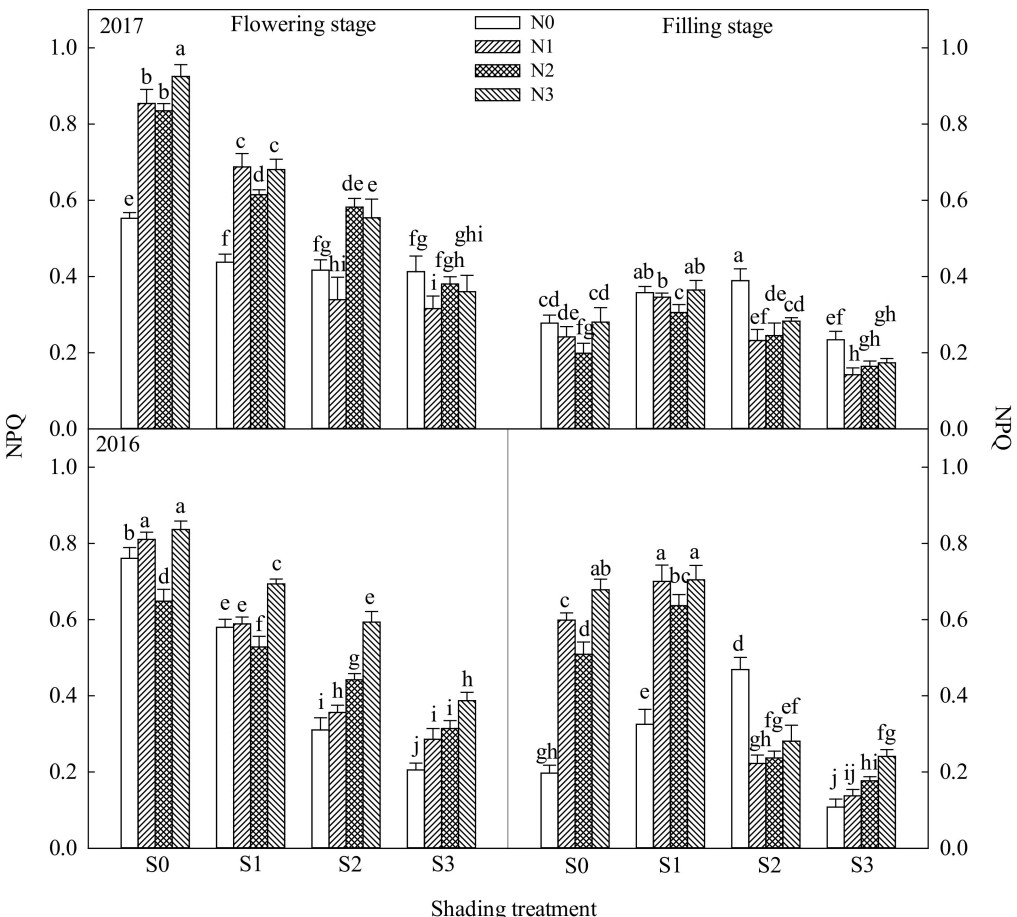

**Figure 9.** Effect of light and nitrogen treatment on non-photochemical quantum coefficient (NPQ) of wheat. Bars indicate SD (n = 3). Values within columns followed by the same letter are statistically insignificant at the 0.05 level.

### 3.5. Dry Matter Accumulation

The experiment showed (Figure 10) that the total above-ground dry matter accumulation (TDA) and the dry matter accumulation of the reproductive organs (DAR) both significantly decreased with the increase in the shade range. Compared with the control (S0), TDA of S1, S2, and S3 treatments were reduced by 1.1–19.1%, 28.0–45.6%, 41.2–53.0%, respectively, and TDA of reproductive organs decreased by 0.5–12.6%, 41.3–52.3%, and 56.1–64.9%, respectively. Under the conditions of S0 and S1, TDA and DAR of N2 treatment were significantly higher than other treatments. TDA of N2 treatment was 7.1–16.2%, 0.6–20.7%, and 56.0–340.8% higher than that of N3, N1, and N0 treatment, respectively. DAR of N2 treatment was 14.0–21.1%, 0.3–12.2%, and 5.8–323.0% higher than that of N3, N1, and N0 treatment, respectively. Under the conditions of S2 and S3, there was no significant difference in TDA and DAR among different nitrogen fertilizer treatments.

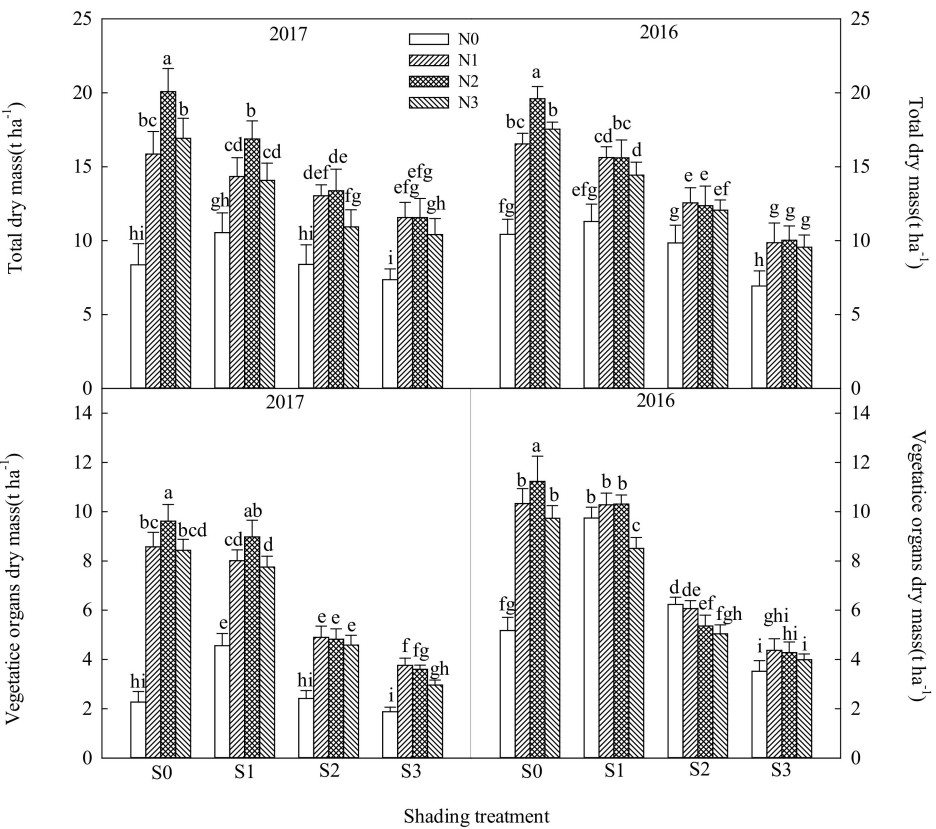

**Figure 10.** Effect of light and nitrogen treatment on above-ground dry matter accumulation and distribution of wheat. Bars indicate SD (n = 3). Values within columns followed by the same letter are statistically insignificant at the 0.05 level.

### 3.6. Yield and Yield Components

The variance analysis showed (Table 3) that the effects shading levels (S), nitrogen levels (N) and their interactions (S × N) on yield were extremety significant.

**Table 3.** Analysis of variance of wheat yield under the treatment of shading and N fertilizer.

| Sources of Variation | d.f. | SS | MS | F |
|---|---|---|---|---|
| Replication | 5 | 1,154,668.92 | 230,933.78 | 0.42 |
| Shade | 3 | 206,345,398.85 | 68,781,799.62 | 126.39 ** |
| Error (S) | 15 | 8,162,533.31 | 544,168.89 | |
| Nitrogen | 3 | 75,528,222.43 | 25,176,074.14 | 81.98 ** |
| S*N | 9 | 45,853,371.71 | 5,094,819.08 | 16.59 ** |
| Error (N) | 60 | 18,424,595.55 | 307,076.59 | |
| Total | 95 | 355,468,790.77 | | |

**—highly significant ($p < 0.01$).

The yield decreased with the increase in shading range (Table 4). The yield of S1, S2, and S3 treatments were 10.9%, 31.4%, and 67.6% lower than that of S0 treatments, respectively. Under the conditions of S0 and S1, the yield increased first and then decreased with the increase in nitrogen application. The yield of N2 treatment was 1.5–12.4%, 12.0–14.8% and 48.3–231.2% higher than that of N3, N1 and N0 treatments, respectively; under the conditions of S2 and S3, the yield decreased with the increase in nitrogen application. There was no significant difference in yield among different nitrogen application treatments, but the yield with nitrogen application treatments was significantly higher than that without nitrogen treatment.

Through analyzing yield components, it was found that the spike number and kernels per ear decreased with the increase in shade range. There was no significant difference in 1000-grain weight between S1 and S0 treatments, but 1000-grain weight of S1 and S0 treatments was significantly higher than that of S2 and S3 treatments. Under the conditions of S1 and S0, the spike number first increased and then decreased with the increase in nitrogen application. The spike number of N2 treatment was the highest. There was no significant difference in the kernels per ear and the 1000-grain weight among different nitrogen treatment; under the conditions of S2 and S3, there was no significant difference in the spike number, kernels per ear, and the 1000-grain weight among different nitrogen treatment, indicating that under moderate and excessive shading conditions, the regulating effect of nitrogen application on the three elements of yield is reduced, and the yield is reduced.

**Table 4.** Change of yield and yield components in the different treatments.

| | Treatment | Spike Number ($10^4$ ha$^{-1}$) | Kernels Per ear | 1000-Grain Weight (g) | Yield (kg ha$^{-1}$) |
|---|---|---|---|---|---|
| S0 | N0 | 446.7 ± 6.3 [gh] | 24.2 ± 1.1 [ef] | 40.4 ± 1.0 [bc] | 2716.5 ± 632.2 [i] |
| | N1 | 659.1 ± 63.2 [b] | 32.2 ± 0.7 [a] | 40.7 ± 2.0 [bc] | 7833.0 ± 397.5 [b] |
| | N2 | 714.4 ± 41.7 [a] | 32.6 ± 1.1 [a] | 41.0 ± 1.0 [bc] | 8997.0 ± 766.5 [a] |
| | N3 | 648.6 ± 44.5 [bc] | 31.3 ± 0.8 [ab] | 40.1 ± 1.0 [c] | 8001.0 ± 89.7 [b] |
| S1 | N0 | 478.9 ± 19.3 [g] | 26.7 ± 0.4 [d] | 44.4 ± 1.5 [a] | 4657.5 ± 902.5 [fg] |
| | N1 | 614.1 ± 9.4 [bcd] | 30.1 ± 0.6 [bc] | 41.2 ± 2.0 [bc] | 6171.0 ± 188.1 [cd] |
| | N2 | 634.2 ± 18.0 [bc] | 31.7 ± 2.2 [ab] | 42.5 ± 0.4 [ab] | 6907.5 ± 279.3 [c] |
| | N3 | 612.1 ± 32.2 [bcd] | 31.9 ± 1.8 [ab] | 42.2 ± 1.7 [ab] | 6805.5 ± 228.7 [c] |
| S2 | N0 | 416.7 ± 13.2 [h] | 25.4 ± 0.5 [def] | 42.2 ± 1.2 [bc] | 3583.5 ± 416.4 [h] |
| | N1 | 602.4 ± 26.2 [cde] | 29.5 ± 0.9 [c] | 36.3 ± 1.0 [d] | 5497.5 ± 1227.0 [de] |
| | N2 | 594.9 ± 36.0 [cde] | 29.4 ± 0.4 [c] | 35.3 ± 1.0 [de] | 5422.5 ± 148.1 [def] |
| | N3 | 562.2 ± 28.2 [def] | 30.3 ± 1.5 [bc] | 34.0 ± 2.0 [e] | 5328.0 ± 212.7 [ef] |
| S3 | N0 | 403.9 ± 23.4 [g] | 23.8 ± 0.6 [f] | 40.8 ± 1.8 [bc] | 2785.5 ± 94.6 [i] |
| | N1 | 553.0 ± 16.5 [ef] | 26.7 ± 0.7 [d] | 29.8 ± 1.7 [f] | 3978.0 ± 326.2 [gh] |
| | N2 | 576.6 ± 24.6 [def] | 25.1 ± 0.5 [def] | 29.7 ± 0.6 [f] | 3736.5 ± 625.3 [h] |
| | N3 | 530.5 ± 13.3 [f] | 25.9 ± 0.3 [de] | 28.4 ± 1.0 [f] | 3637.5 ± 202.5 [h] |

Values are means ± SD (*n* = 3). Values within columns followed by the same letter are statistically insignificant at the 0.05 level.

Correlation analysis showed (Table 5) that the yield was extremely significantly positively correlated with Pn, TDA, DAR, strong correlated with spike number, medium correlation with LAI at flowering stage and kernels per ear, indicating that certain photosynthetic area, photosynthetic capacity, dry matter accumulation and distribution to grains under shading conditions are beneficial to increasing the spike number, and kernels per ear, thereby improving yield.

**Table 5.** Correlation coefficients of yield with photosynthetic parameters, dry matter accumulation and yield components.

Columns 1–9 belong to **Flowering Stage** and columns 10–18 belong to **Filling Stage** (LAI, Pn, gs, Ci, Tr, Fv/Fm, ΦPSII, qP, NPQ).

| The Correlation Coefficient | | LAI | Pn | gs | Ci | Tr | Fv/Fm | ΦPSII | qP | NPQ | LAI | Pn | gs | Ci | Tr | Fv/Fm | ΦPSII | qP | NPQ | TDA | DAR | Spike Number | Kernels Per Ear | 1000-Grain Weight | Yield |
|---|---|---|---|---|---|---|---|---|---|---|---|---|---|---|---|---|---|---|---|---|---|---|---|---|---|
| Flowering stage | LAI | 1 | 0.58 ** | 0.83 ** | 0.23 | 0.78 ** | 0.38 * | −0.15 | −0.18 | 0.35 * | 0.81 ** | 0.75 ** | 0.52 ** | 0.15 | 0.57 ** | 0.23 | −0.32 | −0.31 | 0.53 ** | 0.86 ** | 0.70 ** | 0.91 ** | 0.61 ** | −0.2 | 0.68 ** |
| | Pn | | 1 | 0.65 ** | −0.35 * | 0.71 ** | −0.29 | −0.70 ** | −0.66 ** | 0.73 ** | 0.31 | 0.83 ** | 0.57 ** | −0.16 | 0.61 ** | −0.07 | −0.73 ** | −0.72 ** | 0.58 ** | 0.76 ** | 0.81 ** | 0.64 ** | 0.70 ** | 0.47 ** | 0.84 ** |
| | gs | | | 1 | 0.24 | 0.83 ** | 0.12 | −0.18 | −0.16 | 0.39 * | 0.74 ** | 0.75 ** | 0.70 ** | 0.14 | 0.74 ** | 0.09 | −0.40* | −0.42 * | 0.38 * | 0.77 ** | 0.72 ** | 0.77 ** | 0.69 ** | −0.03 | 0.66 ** |
| | Ci | | | | 1 | 0.17 | 0.39 * | 0.70 ** | 0.65 ** | −0.3 | 0.33 | −0.21 | 0.25 | 0.44 ** | 0.21 | 0.47 ** | 0.61 ** | 0.57 ** | −0.36 * | −0.05 | −0.2 | 0.07 | −0.28 | −0.29 | −0.14 |
| | Tr | | | | | 1 | 0.07 | −0.33 | −0.38 * | 0.55 ** | 0.59 ** | 0.79 ** | 0.62 ** | 0.24 | 0.66 ** | 0.11 | −0.52 ** | −0.55 ** | 0.46 ** | 0.85 ** | 0.84 ** | 0.75 ** | 0.78 ** | 0.13 | 0.81 ** |
| | Fv/Fm | | | | | | 1 | 0.34 | 0.27 | −0.34 | 0.50 ** | 0.08 | −0.02 | 0.29 | −0.11 | 0.21 | 0.31 | 0.32 | 0.05 | 0.15 | −0.03 | 0.34 | 0.08 | −0.66 ** | −0.06 |
| | ΦPSII | | | | | | | 1 | 0.95 ** | −0.67 ** | 0.11 | −0.54 ** | −0.1 | 0.32 | −0.09 | 0.3 | 0.77 ** | 0.76 ** | −0.60 ** | −0.45 ** | −0.58 ** | −0.29 | −0.59 ** | −0.46 ** | −0.57 ** |
| | qP | | | | | | | | 1 | −0.74 ** | 0.15 | −0.51 ** | −0.06 | 0.24 | −0.04 | 0.2 | 0.70 ** | 0.69 ** | −0.61 ** | −0.49 ** | −0.61 ** | −0.31 | −0.58 ** | −0.50 ** | −0.63 ** |
| | NPQ | | | | | | | | | 1 | −0.09 | 0.52 ** | 0.21 | −0.07 | 0.17 | 0.08 | −0.55 ** | −0.57 ** | 0.40 * | 0.65 ** | 0.69 ** | 0.50 ** | 0.48 ** | 0.56 ** | 0.80 ** |
| Filling stage | LAI | | | | | | | | | | 1 | 0.56 ** | 0.49 ** | 0.17 | 0.56 ** | −0.05 | −0.15 | −0.11 | 0.31 | 0.57 ** | 0.45 ** | 0.66 ** | 0.54 ** | −0.43* | 0.31 |
| | Pn | | | | | | | | | | | 1 | 0.61 ** | −0.09 | 0.64 ** | −0.06 | −0.66 ** | −0.67 ** | 0.70 ** | 0.85 ** | 0.86 ** | 0.74 ** | 0.86 ** | 0.18 | 0.82 ** |
| | gs | | | | | | | | | | | | 1 | 0.36 * | 0.85 ** | 0.04 | −0.23 | −0.27 | 0.28 | 0.53 ** | 0.58 ** | 0.46 ** | 0.49 ** | 0.22 | 0.53 ** |
| | Ci | | | | | | | | | | | | | 1 | 0.11 | 0.25 | 0.26 | 0.21 | −0.33 | 0.14 | 0.08 | 0.2 | 0.03 | −0.25 | 0.07 |
| | Tr | | | | | | | | | | | | | | 1 | 0.04 | −0.3 | −0.34 | 0.35 * | 0.53 ** | 0.59 ** | 0.42 * | 0.51 ** | 0.22 | 0.49 ** |
| | Fv/Fm | | | | | | | | | | | | | | | 1 | 0.39 * | 0.34 | −0.18 | 0.12 | −0.05 | 0.21 | −0.34 | 0.02 | 0.16 |
| | ΦPSII | | | | | | | | | | | | | | | | 1 | 0.99 ** | −0.60 ** | −0.57 ** | −0.67 ** | −0.42 * | −0.73 ** | −0.19 | −0.52 ** |
| | qP | | | | | | | | | | | | | | | | | 1 | −0.58 ** | −0.58 ** | −0.58 ** | −0.69 ** | −0.72 ** | −0.22 | −0.55 ** |
| | NPQ | | | | | | | | | | | | | | | | | | 1 | 0.59 ** | 0.66 ** | 0.41 * | 0.62 ** | 0.26 | 0.57 ** |
| TDA | | | | | | | | | | | | | | | | | | | | 1 | 0.93 ** | 0.91 ** | 0.78 ** | 0.08 | 0.90 ** |
| DAR | | | | | | | | | | | | | | | | | | | | | 1 | 0.73 ** | 0.87 ** | 0.31 | 0.90 ** |
| Spike number | | | | | | | | | | | | | | | | | | | | | | 1 | 0.61 ** | −0.2 | 0.75 ** |
| Kernels per ear | | | | | | | | | | | | | | | | | | | | | | | 1 | 0.1 | 0.71 ** |
| 1000-Grain weight | | | | | | | | | | | | | | | | | | | | | | | | 1 | 0.44 * |
| Yield | | | | | | | | | | | | | | | | | | | | | | | | | 1 |

**—highly significant ($p < 0.01$) and *—significant ($0.01 < p < 0.05$), ($n = 32$).

## 4. Discussion

The size of leaf area determines the amount of light interception, which affects the photosynthesis and the accumulation of photosynthate [28]. In this paper, it is found that with the increase in the shade range, LAI first increases and then decreases and the LAI under mild shading (S1) is higher. Under shading conditions, plants can increase the light interception efficiency by increasing the canopy size, such as increasing the leaf area index (LAI) to get more light [29]. Proper nitrogen application can improve the single leaf area and population leaf area index of wheat [30], and the postponing nitrogen fertilizer is beneficial to delaying the reduction in population leaf area in the later growth period of wheat [31]. In this experiment, under the conditions of S0 and S1, N2 is conducive to increasing the LAI of wheat. However, under moderate and excessive shading conditions, increasing nitrogen application is unfavorable for increasing the LAI of wheat. It indicates that under shading conditions, the regulating effect of nitrogen fertilizer on leaf area is reduced, and the higher the shading degree, the lower the effect [17].

The increase in photosynthetic pigment content may also help capture and use light more effectively [32,33]. In this experiment, under mild shading, chlorophyll a, b and a + b increase significantly, and chlorophyll a/b decreases. After shading, the chlorophyll content of wheat flag leaves increases significantly, and the chlorophyll a/b decreases. This is because shading promotes the compensatory synthesis of chlorophyll content, which makes up for the lack of light by increasing chlorophyll content to maintain basic metabolism [17,34]. Meanwhile, after shading, the increase in chlorophyll b is higher than that of chlorophyll a. Wheat leaves increase the absorption of blue-violet light by increasing the relative content of chlorophyll b, thereby enhancing the utilization of low light by plants in weak light [35,36]. Moreover, the chloroplast's ability to reduce 2, 6-dichloroindophenol is enhanced, and the content of LHCP and the activity of photosynthetic phosphorylation are improved [37]. Under moderate and excessive shading conditions, the decrease in chlorophyll content may be related to the insufficiency of light intensity and the obstruction of chlorophyll synthesis [38]. In this study, proper nitrogen application is beneficial to increasing the chlorophyll content of wheat without shading and under mild shading; under moderate and excessive shading, increasing nitrogen is not conducive to the chlorophyll synthesis. Under shading conditions, the effect of increasing nitrogen fertilizer on chlorophyll synthesis is lower than that of normal light. This is due to the reduced light intensity and insufficient assimilation power (ADP and NADPH) after shading, which restricts photosynthetic carbon assimilation and causes carbon and nitrogen metabolism disorders, affecting the accumulation and transportation of photosynthetic substances such as chlorophyll in crops [39,40].

The Pn reflects the photosynthetic capacity of crops and is the key to the yield [41]. Mu et al. [12] believed that the Pn of wheat flag leaves under shading conditions decreased significantly. The photosynthetic rate of wheat leaves increased under mild shading (shading 8% and 15%) and decreased under moderate shading (shading 23%) [34]. In this paper, the Pn, Tr, and gs decrease with the increase in shading range, while the Ci increases, implying that the decline of crop photosynthetic rate under shading is caused by non-stomatal factors [36]. It is believed in some studies that the decrease in photosynthetic rate under low light is mainly caused by the increase in mesophyll resistance [42] and the decrease in photo system activity [12]. In this experiment, under mild shading, proper nitrogen can weaken the negative effect of shading on wheat photosynthesis; under moderate and excessive shading conditions, the regulating effect of nitrogen fertilizer on photosynthetic rate is reduced. Some researchers also believe that under shading conditions, compared with the control (N0), nitrogen treatment (N1, N2) increase 11.5–27.4%, which is higher that of normal treatment (5.5–23.2%), implying that increasing nitrogen fertilizer can effectively alleviate the adverse effects of low light stress on photosynthesis under shading conditions [17].

The dynamic characteristics of chlorophyll fluorescence can systematically reflect the absorption, transmission, dissipation and distribution of light energy by leaves [43].

In this study, with the increase in the shading degree, the ΦPSII and qP of wheat flag leaves increase significantly, and the NPQ decreases significantly, indicating that under shading conditions, the light energy absorbed by the antenna pigment in PSII dissipates less through heat [44], and the limited excited light energy is used more effectively [34]. Increased nitrogen fertilizer is beneficial to enhancing the ability of wheat leaves to capture light energy, improve the light energy conversion efficiency, and enable wheat to use the captured light energy more effectively for photosynthesis [16]. In this study, it is found that proper nitrogen (N2) is beneficial to improving the ΦPSII, qP and reducing the NPQ in wheat under the conditions of no shading and mild shading. Zhang et al. [17] held that the regulating effect of nitrogen fertilizer on qP and NPQ under shading is higher than that of normal light, but its regulating effect on ΦPSII is lower than that under normal light.

Shading reduces dry matter accumulation and distribution to grains of wheat [45], resulting in lower grain yields [13]. The author believes that with the increase in shading degree, TDA above ground of wheat and DAR decrease, and the yield reduces significantly. The reduction in grain yield caused by shading depends on the shading intensity and duration, the growth stage and the characteristics of the plant variety [46]. From jointing stage to maturity stage, medium shading degree (≥85% of total radiation) increases the grain yield of shade-tolerant varieties, but shading degree over 22% significantly reduces yield [12,34]. Shading limits the availability of carbohydrates [47], reduces the speed of spikelet growth, reduces the number of spikelets [48], and also reduces the germination and development of florets [49]. Post-anthesis shading leads to the reduction in wheat yield, which is caused by the decrease in 1000-grain weight and kernels per ear [50,51]. The author in this study holds that the spike number and kernels per ear decrease with the increase in the shading degree. There is no significant difference in 1000-grain weight between S1 treatment and S0 treatment, but 1000-grain weight of S1 treatment is significantly higher than that of S2 and S3 treatments. Increasing nutrient elements is one of the important ways for crops to improve the utilization of low light [52]. The research of Zhang et al. [17] showed that increased nitrogen fertilizer application can alleviate the adverse effects of low-light stress on photosynthesis and realize high yield by increasing nitrogen, but the high yield effect by increasing nitrogen application under low light is relatively low. Under the conditions of no shading and mild shading, the yield first increases and then decreases with the increase in nitrogen application, while the yield decreases with the increase in nitrogen application under moderate and excessive shading conditions. Zhou et al. [53] also believed that without shading, the yield increases with the increase in nitrogen application; under shading conditions, the yield decreases with the increase in nitrogen application. With the increase in shading degree, the yield shows a continuous downward trend. In this experiment, the yield is significantly positively correlated with LAI, Pn, TDA, DAR, spike number, and kernels per ear during flowering stage, implying that under shading conditions, keeping photosynthetic area, photosynthetic capacity, dry matter accumulation and distribution to grains is beneficial to improve the spike number and kernels per ear, thereby improving yield.

## 5. Conclusions

Under mild shading conditions, proper nitrogen application (138.0 kg ha$^{-1}$ nitrogen was applied at the jointing stage) increases the photosynthetic area and pigment content of wheat, the proportion of the ΦPSII of leaf photosynthetic apparatus and open PSII reaction centers increases, and the non-radiative energy dissipation of PSII reaction centers decreases; under moderate and excessive shading conditions, the photosynthetic area and pigment content of wheat decrease. The proportion of the ΦPSII of leaf photosynthetic apparatus PSII and open PSII reaction centers increases. However, it cannot compensate for the yield decrease in wheat caused by significant reduction in the light intercepted by the wheat canopy, and the reduction in photosynthetic rate, the accumulation of photosynthetic matter, and the distribution to the grain. Increased application of nitrogen fertilizer (applying pure nitrogen more than 103.5 kg ha$^{-1}$ at the jointing stage) may reduce or even



inhibit its regulating effect on wheat photosynthetic capacity, material production and yield. Therefore, under jujube-wheat intercropping, and apricot-wheat and walnut-wheat with light shade in the early stage, photosynthetic capacity of wheat leaves and dry matter accumulation and transfer to grains can be regulated by nitrogen fertilizer, which is beneficial to compensating for the effect of insufficient light on the wheat growth of wheat, exert the coupling effect of light and nitrogen, and achieve high yield of wheat; under moderate or excessive shading conditions (apricot-wheat and walnut-wheat in full fruit period), the regulating effect of nitrogen fertilizer on wheat is reduced, so the nitrogen fertilizer should be reduced moderately.

**Author Contributions:** Conceptualization, F.C.; Data curation, H.Z.; Formal analysis, Q.Z. and F.C.; Funding acquisition, Q.Z.; Investigation, H.Z., Z.W., L.W., X.L., Z.F., Y.Z., J.L., X.G. and J.S.; Methodology, F.C.; Project administration, Q.Z.; Writing—original draft, H.Z. and Z.W.; Writing—review and editing, Q.Z. and F.C. All authors have read and agreed to the published version of the manuscript.

**Funding:** This project was supported by the Youth Science and Technology Backbone Innovation Ability Training Project of Xinjiang Academy of Agricultural Sciences (xjnkq-2019013), National Natural Science Foundation of China (31560370), National Key R&D Program of China (2016 YFD0300110), Xinjiang Outstanding Young Scientific and Technological Personnel Training Program (2020 Q009) and Ten-zan Innovation Team Program of Xinjiang (2020 D14001).

**Institutional Review Board Statement:** Not applicable.

**Informed Consent Statement:** Not applicable.

**Data Availability Statement:** Not applicable.

**Conflicts of Interest:** The authors declare no conflict of interest. The funders had no role in the design of the study; in data collection, analyses, or interpretation of data; in the writing of the manuscript, or in the decision to publish the results.

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
