# Peer review of "Effects of Nitrogen Fertilizer on Photosynthetic Characteristics, Biomass, and Yield of Wheat under Different Shading Conditions"

_agronomy, doi:10.3390/agronomy11101989_

Round 1

Reviewer 1 Report

The subject that was studied has a medium importance. Also, shading by fruit trees has significant differences compared with technical shading, e.g., air temperature, air moisture, etc. This manuscript in study only one factor e.g. the shading. However, to have a general conclusion about wheat-fruit tree intercropping, it is necessary to study and the other parameters. Finally, the importance of this manuscript is increased if we consider the increase of the human population, believing that the cultivated area is slightly decreased or not increased.

-I read in the material and methods (lines 123-134) that the experimental design is a split plot with three replications e.g. three blocks? Is it wright?

However, as I can read from the anova table line 366, the d.f. revealed that the design analysis didnot include the block inside. More importantly, the experiment analyzed as a factorial experiment which is wrong and quite different from spilt-split plot.

-Also, in correlation lines 399-406, please discuss only those correlations that are above 0.60. And it is better to say strong correlation when r>0.75-0.80 and medium correlation when r=0.6-0.7.

-The discussion is ok

The conclusion could be enriched e.g. to analyze a little more the lines 520-523 and their effect in increasing agricultural production. “herefore, under early shading conditions, photosynthetic capacity of wheat leaves can be regulated by nitrogen fertilizer, which is beneficial to compensating for the effect of insufficient light on the wheat growth of wheat, exert the coupling effect of light and nitrogen, and achieve high yield of wheat”

Reviewer 2 Report

The manuscript contains interesting results from research on wheat, but requires considerable elaboration and supplementation. There is no description of weather conditions and full characteristics of soil conditions. It is also unclear how many and what years of research the results come from.

Comments:

Abstract is too long, please shorten it

Line 20: was the experience three years?

Line 20: correct wheat variety Xindong 20

Line25: correct  kg·ha-2

Line 26: correct nitrogen).

Line 44: correct Pn – net photosynthesis rate;

Line 44: correct NPQ – quantum

Line 56: correct (Triticum aestivum L.)

Line 118: in the abstract is 2016-2018

Line 119: please indicate soil type according to IUSS Working Group WRB. World Reference Base for Soil Resources 2014. International Soil Classification System for Naming Soils and Creating Legends for Soil Maps. Update 2015;World Soil Resources Raport 106; FAO: Rome, Italy, 2015; 188 p.

Line 119: there is no complete description of soil conditions and description of soil testing methods.

Line 120: please provide the characteristics of the wheat variety

Table 1: there is no reference to the table in the text

Line 134: correct 8 m2,

Line 136: correct 50 cm

Line 141, 142: the description shows that the experience was two years?

Line 142, 143: please specify the composition of the fertilizers

Line 157: correct width × 0.83 m).

Line 208: please provide full details of the software manufacturer

Line 228, 275, 278, 280, 288, 297, 316, 319, 336, 360: please add explanations of letters in the figures

Line 366: correct Table 3. Analysis

Line 396, tabele 4: correct ha

Line 407: correct Table 5. Correlation…, (n =?)

References: please remove publications from before 2000 (5, 12, 27, 33, 43-45, 49, 51)

Please correct any minor errors, such as missing spaces, etc.

Round 2

Reviewer 1 Report

I am happy to read that the majority of proposed changes have been incorporated, and I would like to thank authors for that.

Howerer, in the previous version I informed the authors that the statistical analysis of yield is wrong, e.g. the experimental design is split-splot while the statistical analysis of yield table 3 (line 402) is for factorial design. So it is crucial to change the statistical analysis which could give different results. It is crucial to make this changes if you want to publish your manuscript.

I do not have any comment in the pdf file. In order to help you I have add a picture with the statistical analysis, in the attached Word file 
